# Dual Monitoring of Cracking and Healing in Self-healing Coatings Using Microcapsules Loaded with Two Fluorescent Dyes

**DOI:** 10.3390/molecules24091679

**Published:** 2019-04-30

**Authors:** Young Kyu Song, Tae Hee Lee, Jin Chul Kim, Kyu Cheol Lee, Sang-Ho Lee, Seung Man Noh, Young Il Park

**Affiliations:** 1Research Center for Green Fine Chemicals, Korea Research Institute of Chemical Technology, Ulsan 44412, Korea; yksong@krict.re.kr (Y.K.S.); thlee@krict.re.kr (T.H.L.); jckim81@krict.re.kr (J.C.K.); leekc@krict.re.kr (K.C.L.); slee@krict.re.kr (S.-H.L.); smnoh@krict.re.kr (S.M.N.); 2Department of Chemical Engineering, Ulsan National Institute of Science and Technology, Ulsan 44919, Korea

**Keywords:** extrinsic self-healing, self-healing detection, aggregation-induced emission, dye-loaded microcapsule, self-healing monitoring

## Abstract

We report the development of an extrinsic, self-healing coating system that shows no fluorescence from intact coating, yellowish fluorescence in cracked regions, and greenish fluorescence in healed regions, thus allowing separate monitoring of cracking and healing of coatings. This fluorescence-monitoring self-healing system consisted of a top coating and an epoxy matrix resin containing mixed dye loaded in a single microcapsule. The dye-loaded microcapsules consisted of a poly(urea-formaldehyde) shell encapsulating a healing agent containing methacryloxypropyl-terminated polydimethylsiloxane (MAT-PDMS), styrene, a photo-initiator, and a mixture of two dyes: one that fluoresced only in the solid state (DCM) and a second that fluoresced dramatically in the solid than in the solution state (4-TPAE). A mixture of the healing agent, photo-initiator, and the two dyes was yellow due to fluorescence from DCM. On UV curing of this mixture, however, the color changed from yellow to green, and the fluorescence intensity increased due to fluorescence from 4-TPAE in the solid state. When a self-healing coating embedded with microcapsules containing the DCM/4-TPAE dye mixture was scratched, the damaged region exhibited a yellowish color that changed to green after healing. Thus, the self-healing system reported here allows separate monitoring of cracking and healing based on changes in fluorescence color.

## 1. Introduction

One of the roles of a coating is to protect the underlying substrate from external impacts, thereby protecting it from corrosion and other processes that may compromise its mechanical properties. Thus, coatings can extend the lifetimes of materials and, in various contexts, enhance public safety. The development of self-healing coatings, and techniques for monitoring such coatings in real time, has proved to be an effective means of extending the lifetime of materials. Recently, a technique was introduced for monitoring self-healing coatings based on changes in color or fluorescence at damaged locations on the coating surface. [1,2,3,4,5,6,7,8,9,10] This technique is of great practical utility, as it allows the condition of a coating to be assessed through visual inspection. Nevertheless, most of the recent reported research has been conducted to only detect external impacts or cracks in self-healing coatings. [2,3,4,5,6,7,8,9] However, to fully detect self-healing of a coating surface, a technique for simultaneously detecting cracking and healing is needed [10].

In a previous report, we studied self-healing coatings containing fluorescence dye-loaded microcapsules, in which self-healing gave rise to aggregation-induced emission (AIE) that could be used to detect the healing process (See Figure 1) [1,11]. Furthermore, we studied the detection of cracking and healing separately by adding a fluorescence layer [10]. However, formation of an additional layer in the coating system has the disadvantage of making the process uncomfortable. To properly monitor a self-healing system, cracking and healing should be detected separately, because it is important to maintain the role of the coating by monitoring the coating’s surface. This is especially the case for irreversible, extrinsic self-healing materials using microcapsules.

Here, we introduce a self-healing system that shows different fluorescence color behaviors after cracking and healing, allowing the two processes to be monitored separately. This dual monitoring capability is achieved by loading each microcapsule with two types of fluorescent dye: a normal ACQ (aggregation-caused quenching) dye with yellowish fluorescence and an AIE dye with a highly contrasting greenish fluorescence. These two fluorescent dyes respond differently during cracking and healing of the self-healing coating (see Figure 1). The normal ACQ fluorescent dye maintains its fluorescence intensity on going from the liquid to the solid phase [12,13], whereas the AIE dye exhibits dramatically increased fluorescence in the solid compared to the liquid phase [1,14,15,16,17,18] (see Figure 2a). Thus, if the coating is in its original, intact state, it will show weak fluorescence. On cracking of the coating, however, normal fluorescent dye begins to orange fluoresce at 570 nm, but the AIE dye exhibits relatively little fluorescence. Then, as the coating solidifies during self-healing under UV irradiation, the AIE dye fluoresces more at 440 nm than normal fluorescent dye, causing the color to change from an orange to a greenish fluorescence (see Figure 2b).

## 2. Results and Discussion

### 2.1. Fluorescence Properties of the Healing Agent following Photo-Curing

To examine the change in fluorescence color of the dyes during self-healing, a sample comprised of a liquid photo-curable healing agent containing the normal dye, AIE dye, or both dyes was subjected to photo-curing under UV (wavelength, 365 nm) irradiation, and the photo-polymerization was monitored by rheometry (Figure 3). In the case of irradiation of the healing agent containing a photo-initiator, the storage modulus (G’) began to increase after about 1200 s of UV light irradiation, and the modulus increased to 10^8^ Pa by 3600 s. In contrast, the healing agent without a photo-initiator showed no change in G’ during UV irradiation, indicating that polymerization did not occur.

When the healing agent containing the photo-initiator and only the normal dye ((4-(dicyanomethylene)-2-methyl-6-(4-dimethylaminostyryl)-4*H*-pyran, DCM) was irradiated with UV light, fluorescence color and intensity remained relatively unchanged after 1800 s (see Figure 3a). In the case of the healing agent containing the photo-initiator and only the AIE dye (1,1,2,2,-tetrakis(4-(diphenylamino)phenyl)ethane, 4-TPAE) (Figure 3b), the fluorescence color changed from violet (UV source color) to blue, and the intensity dramatically increased after 3000 s, which the modulus became high. Finally, when the healing agent containing photo-initiator and a mixture of the normal and AIE dyes was subjected to UV irradiation (Figure 3c), the fluorescence color changed from yellow to green, and the intensity increased following photo-curing. These changes in fluorescence color and intensity during curing in the rheometer could also be observed with the naked eye (see Appendix A). Thus, these findings show that when the healing agent containing both dyes underwent photo-curing under UV light, the fluorescence emission changed from yellow (fluorescence of only the normal dye (DCM) in the liquid state) to a green due to fluorescence from both the normal and AIE dyes (4-TPAE) in the solid state. Collectively, the above results indicate that if microcapsules containing both the normal and AIE dyes are included in a self-healing coating, cracking of the coating will give rise to yellow emission from the normal dye, but as healing progresses, the color will change to green as emission from the AIE dye begins. Thus, this system was capable of monitoring cracking and healing separately based on changes in fluorescence color.

### 2.2. The Self-Healing Coating System with Dye-Loaded Microcapsules

Dye-loaded microcapsules containing the healing agent (MAT-PDMS and styrene), photo-initiator (Benzoin isobutyl ether, BIE), and fluorescent dyes (DCM and 4-TPAE) with a poly(urea-formaldehyde) (UF) shell were prepared by in situ polymerization in an oil-in-water emulsion (Figure 4a). The synthesized microcapsules were all spherical particles with a smooth outer surface (Figure 4b–d), diameters of 20 to 260 μm, and an average size of about 130 μm (Appendix A). The Fourier transform (FT)-IR spectrum of the microcapsules (Appendix A) contained several absorption bands that could be attributed to the poly(urea-formaldehyde) shell, including features at 3730–3030 cm^−1^ corresponding to N–H and O–H stretching vibrations, 1643 cm^−1^ corresponding to the C=O stretching vibration, and 1564 cm^−1^ corresponding to the C–N stretching vibration. The FT-IR spectrum of the core material, which served as a healing agent, included C=O stretching and C=C stretching absorption bands at 1721 cm^−1^ and 1639 cm^−1^, respectively. These results indicated that the synthesized microcapsules were well-formed, with the healing agent encapsulated by the UF shell [1,5].

To examine the change in fluorescence of the microcapsules during photo-curing, microcapsules containing the healing agent plus DCM, 4-TPAE, or both dyes were treated with 365 nm UV light for 1 h [19]. As shown Figure 4a, the DCM-loaded microcapsule showed yellow fluorescence before irradiation and yellow fluorescence of similar intensity after irradiation (see Figure 4(b-1,b-2)). By contrast, the 4-TPAE-loaded microcapsules showed no fluorescence before irradiation but strong blue fluorescence after irradiation (Figure 4(c-1,c-2)). Lastly, microcapsules loaded with both dyes showed yellow fluorescence before irradiation but green fluorescence after irradiation (Figure 4(d-1,d-2)). These results confirmed the suitability of the dye-loaded microcapsules for cracking and healing detection in self-healing coatings.

### 2.3. Dual Monitoring of Cracked and Healed Regions with Fluorescent, Dye-Loaded Microcapsules

To test the suitability of the self-healing system for separate detection of cracking and healing by monitoring fluorescence, self-healing coatings comprised of an epoxy matrix loaded with microcapsules containing a healing agent and a mixture of DCM and 4-TPAE dyes were prepared (Appendix A). The coating surface was scratched with a razor and then irradiated with UV light. Optical microscope images of scratched surfaces of coatings containing microcapsules loaded with DCM, 4-TPAE, and a DCM/4-TPAE mixture are shown in Figure 5.

As expected, the scratched control coating without microcapsules showed no fluorescence before and after irradiation (Figure 5a). When DCM-loaded microcapsules were included in the coating, yellowish fluorescence was observed in both the cracked and healed states, with little difference in fluorescence color or intensity between the two states. In the case of the self-healing coating system with 4-TPAE dye-loaded microcapsules, however, no fluorescence was detected along the scratch, but blueish fluorescence emerged during healing (Figure 5(c,c-1)). Lastly, the scratched self-healing coating containing microcapsules with both the DCM and 4-TPAE dyes showed yellowish fluorescence along the scratch that changed to greenish fluorescence during healing. These results showed that, when microcapsules containing both dyes were included in a self-healing coating, the contrasting fluorescence properties of the dyes could be exploited to distinguish scratched and healed regions.

## 3. Experimental Parameters

### 3.1. Materials

Urea, an aqueous formaldehyde solution (37 wt%), poly(ethylene-*alt*-maleic anhydride) (EMA), resorcinol, benzoin isobutyl ether (BIE), styrene, 1-octanol, and 4-(Dicyanomethylene)-2-methyl-6-(4-dimethylaminostyryl)-4*H*-pyran (DCM) were purchased from Sigma-Aldrich. Methacryloxypropyl-terminated polydimethylsiloxane (MAT-PDMS) (molecular weight 380–550 g/mol, viscosity 4–6 cSt) was acquired from Gelest. Ammonium chloride was supplied from Duksan Pharmaceutical. The coatings were applied using an epoxy polymeric resin (YD-114), based on bisphenol-A, diluted with aliphatic glycidyl ether and a cycloaliphatic amine-modified curing agent (KH-816) for the epoxy resin, which had been kindly donated by Kukdo Chemicals. An epoxy paint for the top coating containing gray pigment was kindly donated by Samjoong Inc. All chemicals and organic solvents were used with purification, with the exception of styrene, which was passed over a basic alumina column to remove any inhibitors.

### 3.2. Instruments

Proton nuclear magnetic resonance (^1^H NMR) spectra were taken on a Bruker Ultrasheild 300 MHz spectrometer in deuteriochloroform (CDCl_3_). IR spectra were recorded on a Fourier transform infrared (FT-IR) spectrophotometer (Nicolet 6700, Thermo Scientific, Waltham, MA, USA). UV-Vis spectroscopy was conducted using a Lambda 25 UV-Vis spectrometer (Perkin-Elmer). Fluorescence spectra were collected using a Varian Cary Eclipse fluorescence spectrometer. Rheological experiments during photocuring were carried out by connecting a rotational rheometer (MARS III, Haake, Thermo Scientific) with and Omni Cure Series 2000 UV-vis mercury lamp curing system (Lumen Dynamics). Photo-irradiation for fluorescence detection and UV curing was conducted using an exposure system (EXECURE 4000, HOYA, Tokyo, Janpan) equipped with a mercury lamp. A mechanical stirrer (EUROSTAR 20, IKA) was used for microencapsulation. A microscope (BX-53, Olympus) was used to take pictures of the damaged surface, healed surface, and microcapsules. The microcapsule size was analyzed using a microscope equipped with a Charge-coupled device (CCD) camera (DP73, Olympus) and image analysis software (cellSens Entry, Olympus, Tokyo, Japan). The mean diameter was determined from a data set of at least 300 individual diameter measurements. A scanning electron microscope (SEM) (SNE-3000M, SEC) was used to examine morphology, microcapsule shape, and coating surface.

### 3.3. Synthesis of Dye-Loaded Microcapsules

A 2.5 wt% aqueous solution of EMA (5 mL) was added to distilled water (20 mL), to which urea (0.504 g), resorcinol (0.050 g), and ammonium chloride (0.050 g) were then added while stirring. The pH of the resultant solution was adjusted to 3.5 using a 10% NaOH solution. One drop of 1-octanol was then added to this solution to eliminate surface bubbles. The resultant mixture was agitated at 1200 rpm, and to the stirred solution 10 mL of the core material was added, which consisted of MAT-PDMS, styrene, BIE, 4-(dicyanomethylene)-2-methyl-6-(4-dimethylaminostyryl)-4*H*-pyran (DCM), and (1,1,2,2,-tetrakis(4-(diphenylamino)phenyl)ethane (4-TPAE) as AIE luminogen (the content of the core material is shown in Section 3.3). 4-TPAE was synthesized according to the method report in the literature [15]. To the agitated emulsion, a 37 wt% formaldehyde (1.456 g, 0.0179 mol) solution was added. The temperature of the resulting mixture was raised to 60 °C, and the solution was heated at that temperature for 5.5 h. The reaction mixture was cooled to room temperature, and the microcapsules were separated using vacuum filtration. The microcapsules were washed with water and acetone and then air-dried. The yields of the microcapsule were approximately 80%. Microcapsules without 4-TPAE were synthesized using the same process as in the above-mentioned procedure with 4-TPAE. The microcapsule and a filter paper were placed between two glass slides and pressed together. The cured material was extracted from the squeezed microcapsule. The broken shell materials were washed with chloroform and then dried in an oven at 60 °C. The extracted core and dried shell material were together mixed with KBr to form a pellet, from which IR spectra were acquired.

### 3.4. Preparation of the Self-Healing Coating

DCM dye containing the healing agent (D-HA)-loaded microcapsule, 4-TPAE containing the healing agent (T-HA)-loaded microcapsule, and (DCM and 4-TPAE-containing healing agent) DT-HA-loaded microcapsules were added to the epoxy resin in an amount of 25 wt%, respectively, to form a self-healing coating. A control coating was prepared with epoxy resin without the microcapsule. The coating was applied to the surface of a glass slide and dried for 2 h at 60 °C. The resultant coatings were covered with a grey pigment epoxy coating to block UV light. The microcapsule-containing layer was 400 μm thick, and the top coating layer was 200 μm thick.

## 4. Conclusions

The present results demonstrate that cracking and healing in extrinsic, self-healing coatings can be separately monitored by loading the microcapsules with two dyes, a normal fluorescent dye (DCM) that produces yellowish fluorescence in both the solid and liquid states and an AIE dye (4-TPAE) that shows very weak fluorescence in the liquid state but strong blueish fluorescence in the solid state. To examine the change in fluorescence color of the dyes during self-healing, samples comprised of a liquid photo-curable healing agent containing the normal dye, AIE dye, or both dyes were subjected to photo-curing under UV irradiation, and it was monitored by rheometry. When a coating containing microcapsules loaded with this combination of dyes was scratched, the damaged area appeared yellow due to fluorescence from the normal dye. Following irradiation, however, the healing agent in the microcapsules solidified, causing blue AIE fluorescence to occur in the healed regions. This self-healing coating system containing dye-loaded microcapsules reported here can, thus, be used for dual monitoring of cracking and healing at the same time. 

## Figures and Tables

**Figure 1 molecules-24-01679-f001:**
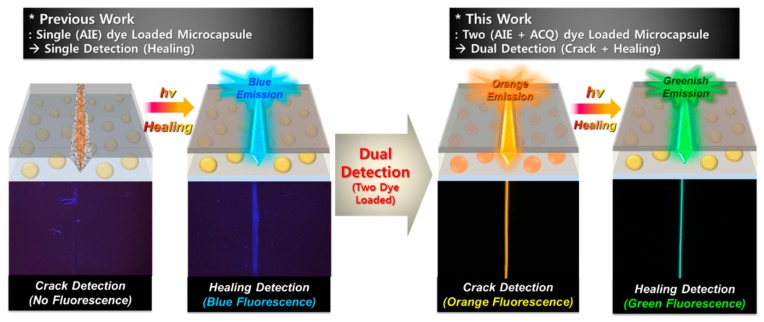
Schematic diagram of dual monitoring of cracking and healing using microcapsules loaded with two fluorescent dyes.

**Figure 2 molecules-24-01679-f002:**
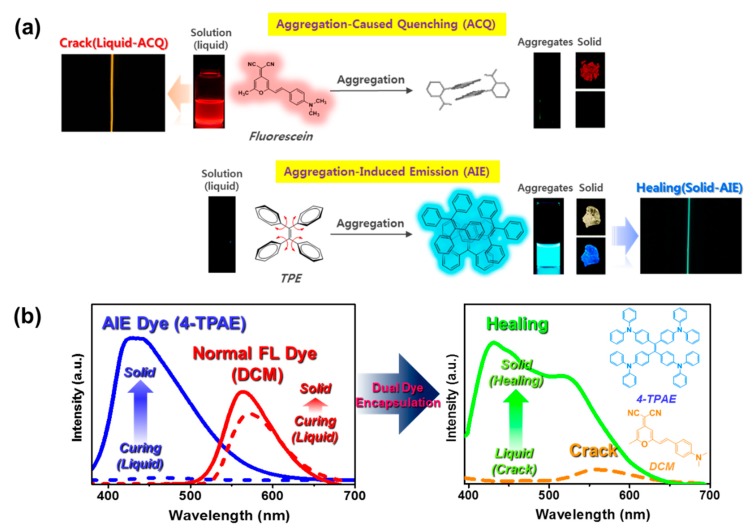
(**a**) Schematic of cracking (aggregation-caused quenching, ACQ) and healing (aggregated-induced emission, AIE) detection using microcapsules loaded with dyes. (**b**) Fluorescence spectrum at 365 nm excitation of the AIE dye (4-TPAE), normal dye (DCM), and 4-TPAE/DCM mixture before and after curing.

**Figure 3 molecules-24-01679-f003:**
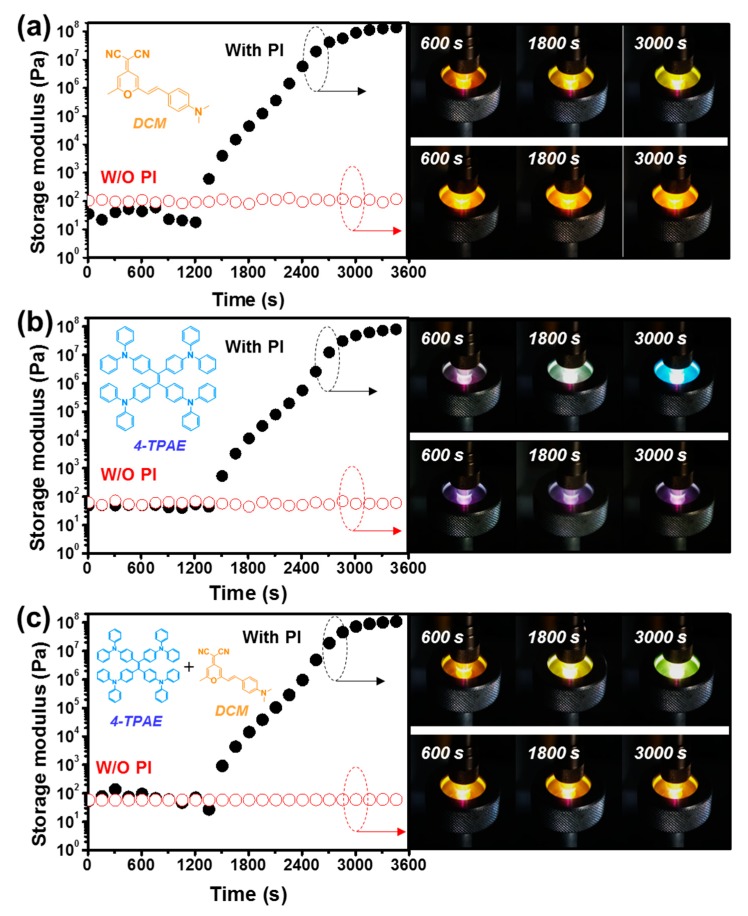
Photo-crosslinking of the healing agents was monitored using a rheology test at 1 Hz frequency under UV irradiation (black dots: healing agent with photo-initiator (PI); red dots: healing agent without photo-initiator). (**a**) DCM, (**b**) 4-TPAE, and (**c**) DCM/4-TPAE mixture in the healing agent.

**Figure 4 molecules-24-01679-f004:**
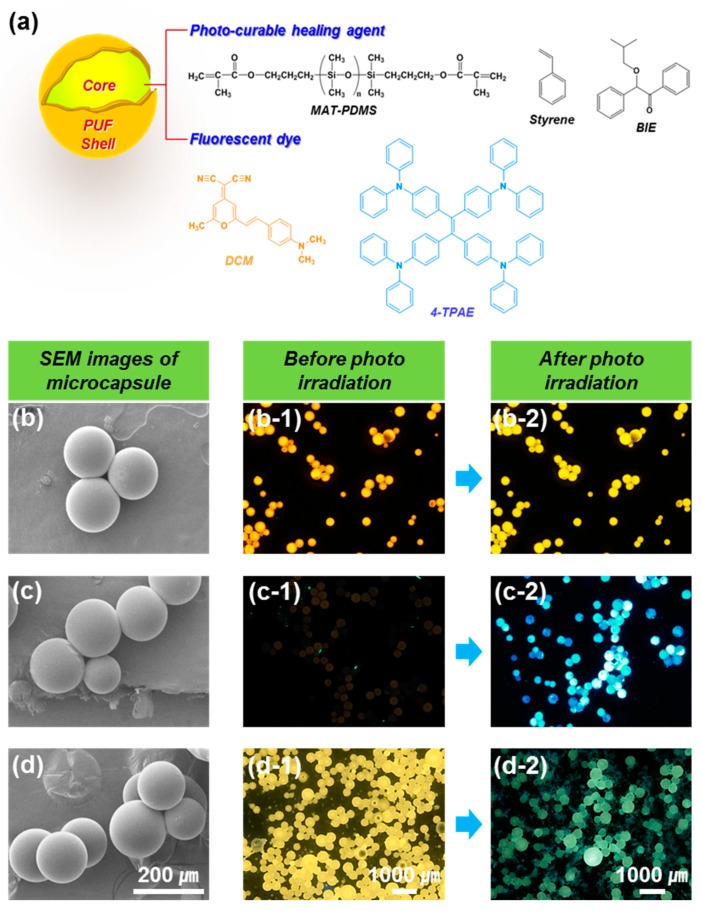
(**a**) Dye-loaded microcapsules and chemical structure of healing agents and fluorescence dyes, SEM, and optical microscope images (under 330 ~ 385 nm UV source) of microcapsules before and after photo-irradiation for (**b**,**b-1**,**b-2**) DCM-loaded, (**c**,**c-1**,**c-2**) 4-TPAE-loaded, and (**d**,**d-1**,**d-2**) 4-TPAE/DCM mixture-loaded microcapsules.

**Figure 5 molecules-24-01679-f005:**
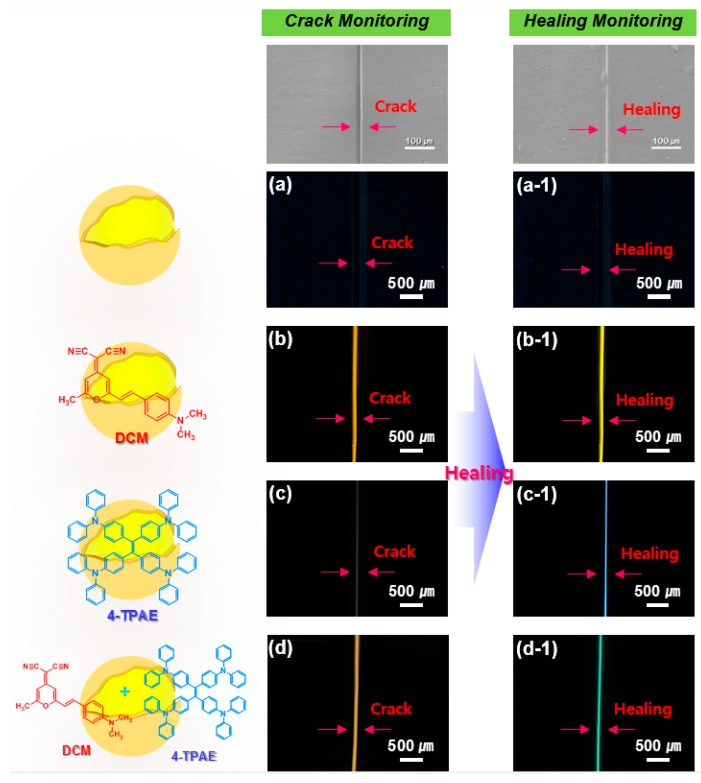
Photographs under 330–385 nm UV source of scratched and healing regions of (**a**,**a-1**) a control coating without microcapsules, with (**b**,**b-1**) DCM-loaded microcapsules, with (**c**,**c-1**) 4-TPAE-loaded microcapsules, and with (**d**,**d-1**) 4-TPAE/DCM mixture-loaded microcapsules in self-healing coatings.

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
