# Peer review of "Dual Monitoring of Cracking and Healing in Self-healing Coatings Using Microcapsules Loaded with Two Fluorescent Dyes"

_molecules, 2019, doi:10.3390/molecules24091679_

Round 1
Reviewer 1 Report
The paper describes the development of self-healing coating systems using microcapsule containing two fluorescence dyes, being applicable to dual monitoring of cracking and healing. The system seemed to be interesting, however a few questions remained. The authors are requested to consider the following points and revised the manuscript.
(1) The authors mentioned that DCM is the ACQ dye. However, the emission of DCM was not completely disappeared after healing. Why?
(2) In the present study, authors developed microcapsule including both DCM and 4-TPAE dyes. However, it is expected that the mixture of DCM-containing capsules and 4-TPAE-containing ones also works similarly as the dual monitoring system. Authors should better describe the merit of microcapsule including both dyes in it.
Reviewer 2 Report
This paper reported monitoring of cracks and healing of materials using microcapsules. Two dyes including DCM and TPAE, together with monomer and iniator, were loaded in polymer microcapsules. The mixture was yellow due to fluorescence from DCM. After curing the mixture, the color changed to green mainly from TPAE. The color change allowed monitoring the cracks and healing of materials. This paper is interesting. Questions are listed as follows:
1. Fig.2 b, Excitation wavelength should be presented.
2. Before curing, the dyes were soluble in monomer. After curing, how did the dyes disperse in materials? They were homogenously dispersed, or formed aggregates? There are interaction between DCM and TPAE in cured materials?
In summary, I recommend its publication after minor revision.
Author Response
1. Fig.2 b, Excitation wavelength should be presented.
--> According to reviewer’s comment, we added excitation wavelength in Figure 2 (b).
2. Before curing, the dyes were soluble in monomer. After curing, how did the dyes disperse in materials? They were homogenously dispersed, or formed aggregates? There are interaction between DCM and TPAE in cured materials?
--> When we checked solubility of the DCM and TPAE in monomer, these dyes were soluble in monomer. And the concentration’s dyes in monomer was very low. Therefore we believed that the dyes were homogenous dispersed and no interaction in cured materials.
In summary, I recommend its publication after minor revision.
--> Thank you for your valuable comments.
Reviewer 3 Report
review report
title:
Dual Monitoring of Cracking and Healing in Self healing Coatings using Microcapsules Loaded with Two Fluorescent Dyes
There are some points of interest in this manuscript. However I believe that significant improvements are needed.
1. The redaction of abstract section should be improved
2. Authors should include more details of the art state for monitoring and healing cracking, highlighting the problematic and opportunities in this scientific area.
3. Discussion section should be improved including a comparison with other materials reported for the detection and healing of cracking, highligthing the advantages for the material reported here
4. the authors must include a detailed explanation for the increase in the luminescence of the material and also for the changes observed in the shape of luminescence band
5. homogenize the format for references
Author Response
There are some points of interest in this manuscript. However I believe that significant improvements are needed.
1. The redaction of abstract section should be improved
--> Thank you for your comment. According to reviewer’s comment, we corrected abstract section.
2. Authors should include more details of the art state for monitoring and healing cracking, highlighting the problematic and opportunities in this scientific area.
--> Thank you for your comment. According to reviewer’s comment, we collected sentences in introduction section as follows.
[Line 39 - 42, page 1]
Nevertheless, most of the recent reported research have been mostly conducted to only detect external impacts or cracks in self-healing coating.[2-9] However, to fully detect of the self-healing coating surface, a techniques for detecting cracks and healing simultaneously are needed.[10]
3. Discussion section should be improved including a comparison with other materials reported for the detection and healing of cracking, highligthing the advantages for the material reported here
--> Thank you for your comment. The advantage of our research is detecting of cracks and healing simultaneously on self-healing coating compare with other research that is detection only crack. We added the highlighting this advantage in introduction section.
4. the authors must include a detailed explanation for the increase in the luminescence of the material and also for the changes observed in the shape of luminescence band
--> Thank you for your comment. According to reviewer’s comment, we collected sentences around maximum wavelength and fluorescent color changes as follows.
[Line 59 - 63, page 2]
Thus, if the coating is in its original intact state, it will show week fluorescence. On cracking of the coating, however, the normal fluorescent dye begins to orange fluorescence at 570 nm but the AIE dye exhibits relatively little fluorescence. Then, as the coating solidifies during self-healing under UV irradiation, the AIE dye more begins to fluoresce at 440 nm than normal fluorescent dye, causing the color to change from orange to greenish fluorescence (see Figure 2(b)).
5. homogenize the format for references
--> Thank you for your comment. According to reviewer’s comment, the format of references was collected.
Reviewer 4 Report
In this work the authors develop self-healing coatings which exhibit fluorescence depending on the state of a crack. Thus sensing and healing can be realized in the same sample. Experiments were performed with two different dyes, where the colors of fluorescence used for detection was shown to be changed because of aggregation of molecules. Interesting characterization is provided in this work, which can be considered for publication after answering the following comments.
- Figure 1 mentions previous work for comparison purposes. The concept of light regulated healing process from microcapsules has been shown previously (ACS Nano 2009, 3, 1753-1760), but, somewhat surprisingly, it is not discussed in this manuscript.
- Also the action of organic molecules (J. Mater. Chem. 2009, 19, 2226-2233) and different spectral regions (for example, UV: Macromol. Chem. and Phys. 2017, 218, 1700213) were introduced earlier. These works have direct link with the main subject of this work and should be mentioned here as well.
- Would pulse or continuous wave irradiation affect the samples differently? This material was also discussed (Colloid. J. 2016, 78, 181-188).
- Figure 4: are those real colors? It is not easy to find details of experiments. Very short figure captions do not allow to obtain sufficient information.
- Figure 5: Figure caption to this figure is quite scarce and one can see a clear difference in color in the last line. Is it possible to extend figure caption?
- The authors discuss using capsules as sensors of mechanical action. Such a concept was discussed for cells (Small 2010, 6, 2858-2862). It is worth mentioning this here.
- Methods are not always fully described. For example, microscope is mentioned but was that an inverted or upright microscope? And how illumination was selected, in reflection?
- Rheology is investigated as a function of frequencies. But it is difficult to find full description here.
- Conclusions can be somewhat extended.
Author Response
In this work the authors develop self-healing coatings which exhibit fluorescence depending on the state of a crack. Thus sensing and healing can be realized in the same sample. Experiments were performed with two different dyes, where the colors of fluorescence used for detection was shown to be changed because of aggregation of molecules. Interesting characterization is provided in this work, which can be considered for publication after answering the following comments.
1) Figure 1 mentions previous work for comparison purposes. The concept of light regulated healing process from microcapsules has been shown previously (ACS Nano 2009, 3, 1753-1760), but, somewhat surprisingly, it is not discussed in this manuscript.
--> Thank you for your comment. According to reviewer’s comment, the reference was added.
2) Also the action of organic molecules (J. Mater. Chem. 2009, 19, 2226-2233) and different spectral regions (for example, UV: Macromol. Chem. and Phys. 2017, 218, 1700213) were introduced earlier. These works have direct link with the main subject of this work and should be mentioned here as well.
--> According to reviewer’s comment, the related reference was added.
3) Would pulse or continuous wave irradiation affect the samples differently? This material was also discussed (Colloid. J. 2016, 78, 181-188).
--> Thank you for your comment. A continuous wave irradiation source was used to cure. Therefore, the mentioned paper is not related to our research.
4) Figure 4: are those real colors? It is not easy to find details of experiments. Very short figure captions do not allow to obtain sufficient information.
--> The optical microscope images of microcapsules in Figure 4 was obtained under UV source (330 ~ 385 nm). So we corrected caption of Figure 4.
5) Figure 5: Figure caption to this figure is quite scarce and one can see a clear difference in color in the last line. Is it possible to extend figure caption?
--> The color difference of in the last line Figure 5 and Figure caption were corrected.
6) The authors discuss using capsules as sensors of mechanical action. Such a concept was discussed for cells (Small 2010, 6, 2858-2862). It is worth mentioning this here.
--> The paper was added as reference on related sentence.
7) Methods are not always fully described. For example, microscope is mentioned but was that an inverted or upright microscope? And how illumination was selected, in reflection?
--> Thank you for your comment. We obtained images using upright microscope. The microscope is equipped with mercury lamp and an optical filter. The coated surfaces were obtained by photo-irradiation with a wavelength range of 330-385 nm through the optical filter.
8) Rheology is investigated as a function of frequencies. But it is difficult to find full description here.
--> Thank you for your comment. We used a rheometer for check the curing behavior of the healing agent containing fluorescent dyes. The frequency during curing was kept to 1 Hz. It was added caption of Figure 3
9) Conclusions can be somewhat extended.
-->Thank you for your comment. Conclusion in manuscript was somewhat extended.